# Methylation Risk Score Modelling in Endometriosis: Evidence for Non-Genetic DNA Methylation Effects in a Case–Control Study

**DOI:** 10.3390/ijms26083760

**Published:** 2025-04-16

**Authors:** Li Ying Thong, Allan F. McRae, Marina Sirota, Linda Giudice, Grant W. Montgomery, Sally Mortlock

**Affiliations:** 1Institute for Molecular Bioscience, University of Queensland, Brisbane, QLD 4072, Australia; l.thong@uq.edu.au (L.Y.T.); a.mcrae@imb.uq.edu.au (A.F.M.); g.montgomery1@uq.edu.au (G.W.M.); 2Bakar Computational Health Sciences Institute, University of California San Francisco, San Francisco, CA 94158, USA; marina.sirota@ucsf.edu; 3Department of Pediatrics, Division of Neonatology, University of California San Francisco, San Francisco, CA 94143, USA; 4Center for Reproductive Sciences, Department of Obstetrics, Gynecology & Reproductive Sciences, University of California San Francisco, San Francisco, CA 94143, USA; linda.giudice@ucsf.edu; 5Australian Women and Girls’ Health Research Centre, University of Queensland, Brisbane, QLD 4006, Australia

**Keywords:** endometriosis, DNA methylation, endometrium, methylation risk score, polygenic risk score

## Abstract

Endometriosis is a chronic gynaecological disease characterised by endometrial-like tissue found external to the uterus. While several studies have reported strong evidence of a genetic contribution to the disease, studies on the environmental impact on endometriosis are limited. DNA methylation (DNAm) can be influenced by genetic and environmental factors and serves as a useful biological marker of the effects of genetic and environmental exposures on complex diseases. This study aims to develop a methylation risk score (MRS) for endometriosis to increase the power to detect DNAm signals associated with the disease and enhance our understanding of the pathogenesis of the disease. Endometrial methylation and genotype data from 318 controls and 590 cases were analysed. MRSs were developed using several different models. MRS performances were evaluated by splitting samples into training and test sets based on independent cohort institutions, and the area under the receiver-operator curve (AUC) was calculated. The maximum AUC obtained from the best-performing MRS is 0.6748, derived from 746 DNAm sites. The classification performance of MRS and polygenic risk score (PRS) combined was consistently higher than PRS alone. This study demonstrates that there are DNAm signals independent of common genetic variants associated with endometriosis.

## 1. Introduction

Endometriosis is a chronic gynaecological disease characterised by endometrial-like tissue found external to the uterus [1]. It is estimated that 6–11% of reproductive-aged women worldwide have been affected by endometriosis [2]. Patients can experience symptoms of pelvic pain and/or infertility, which negatively impact their daily lives, personal relationships, and livelihood, leading to a significant burden on the economy [1]. Estimates of heritability from studies on twin samples have shown that genetic factors contribute approximately 50% of the variation in endometriosis, while the remaining is associated with environmental factors [3,4]. While several studies have explored the link between common genetic variants and endometriosis, providing compelling evidence of a genetic contribution to the disease, studies on the environmental impact on endometriosis have been limited [5,6].

DNA methylation (DNAm) is known to be influenced by both genetic and environmental factors and, therefore, may be a useful biological marker and mediator of the effects of genetic and environmental exposures on complex diseases [7,8]. Evidence of the genetic influence on DNAm patterns is reflected in the estimated heritability of DNAm, which ranges from 0.1 to 0.3, and the discovery of genetic variants that affect DNAm levels, known as DNA methylation quantitative trait loci (mQTLs) [9,10]. Studies have also reported associations between various environmental exposures and changes in DNAm patterns, including effects of socioeconomic status [11], early life environment [12], traumatic events [13], pollutants [14], nutrition [15], and physical activity [16,17].

A link between DNAm and endometriosis has been previously reported in several studies, including hypermethylation of the *HOXA10* and progesterone receptor-B (PR-B) promoters in the endometrium of women with endometriosis [18,19,20,21], the latter providing a possible explanation for progesterone resistance and the reduction in the level of PR-B in endometriosis [19,20,21]. The promoter for the transcription factor for oestrogen biosynthesis, called steroidogenic factor-1 and oestrogen receptor-beta, has been reported to be hypomethylated in endometriotic cells [22,23]. Recently, with the emergence of genome-wide methylation technology, studies have shown DNAm sites mapped to 10 genes correlated to the change in gene expression in the endometrium, contributing to the development or progression of endometriosis [24]. Dyson et al. [25] also identified 403 genes with significantly different methylation patterns between healthy human endometrial and endometriotic stromal cells. However, due to the low sample size of these studies, further studies are needed to verify the results.

A recent study investigating the relationship between DNAm profiles in endometrium of patients with endometriosis and controls without disease identified a significant difference in DNAm profile between stage III/IV endometriosis and controls and networks of methylation sites associated with disease risk [26]. Notably, the variance in endometriosis captured by DNAm was estimated to be 15.4%, and methylation differences between cases and controls were reported to be in the range of 0.03 to 0.08, suggesting that many methylation signals with small effects may contribute to disease. Based on the study’s estimated effect sizes, the study had limited power to reliably detect differences in individual methylation signals between cases and controls [26].

Methylation risk score (MRS) is a numerical value that quantifies an individual’s risk for a particular disease or trait based on their DNAm profile. Applications of MRS include studying the associations between DNAm and a phenotype [27], identifying biomarkers for environmental exposures [28,29], interaction analyses [30,31,32], mediation analyses [27], and predicting the risk of an individual contracting a disease or treatment outcomes [33,34]. MRS is particularly useful for detecting associations between multiple single DNAm sites and a trait, especially when there is insufficient power to achieve statistical significance for individual loci [35]. Several studies have leveraged MRS to validate findings from methylome-wide association studies, including for type 2 diabetes [36] and neurodegenerative disorders, including Alzheimer’s disease, amyotrophic lateral sclerosis, and Parkinson’s disease [37].

We aimed to develop an MRS for endometriosis using endometrial methylation data from 1074 individuals to detect DNAm signals associated with endometriosis and investigate the unique non-genetic contribution of the DNAm to endometriosis risk. We hypothesise that DNAm contributes to endometriosis risk independently of genetic variation and that an MRS derived from endometrial methylation data can effectively identify DNAm signals associated with the disease.

This study provides additional evidence that DNAm, independent of common genetic variants, is associated with endometriosis and emphasises the need for future research with larger sample sizes to explore this relationship further.

## 2. Results

### 2.1. Factors Contributing to Variation in Endometrial DNAm

To identify and address confounders that contribute to variation in endometrial DNAm, a total of 908 samples retained following methylation quality control filtering were included in statistical tests to identify any associations between the potential covariates with endometriosis status and DNAm principal components (PCs). The results showed that age, the institution where the samples were processed, and genetic ancestry were significantly associated with endometriosis status (*p*-value < 0.05) (Table 1). Institutions were also significantly associated with all top 15 DNAm PCs (Appendix A). Furthermore, when plotting PCs, individuals were partially clustered according to institution, as shown in Appendix A. Thus, age and institution were used as covariates during the subsequent analyses. Additionally, genetic PCs were included as covariates during the development of MRS to account for the difference in genetic ancestry and population structure. Surrogate variable (SV) analysis was also conducted to remove batch effects and any hidden sources of variation that were not accounted for by the selected covariates.

### 2.2. Variation in Endometriosis Status Captured by DNAm in Endometrium Independent of Common Genetic Variants

To identify whether DNAm contributes to the variation in endometriosis status seen among participants, we estimated the proportion of variance in endometriosis status that can be captured by DNAm using omics residual maximum likelihood analyses (OREML), a residual maximum likelihood analysis that calculates the estimates from an omics relationship matrix (ORM) generated from DNAm beta values of the endometrium tissue samples. Additionally, to obtain estimates of the proportion of variance in endometriosis status captured by DNAm that was independent of common genetic variants, we simultaneously included the genomic relationship matrix (GRM) and ORM into the OREML model. Estimates from models with and without covariates were recorded in Table 2 to demonstrate the impacts of covariates on the relationship between endometriosis and DNAm in the endometrium.

In the absence of covariates, the proportion of variance captured by both DNAm (12.35%) and common genetic variants (22.38%) (model 3 = 34.73%) was higher than the variance captured by DNAm (model 1 = 19.58%) and common genetic variance (model 2 = 28.83%) alone (Table 2). The proportion of endometriosis variance captured by DNAm in model 3 changes slightly when covariates such as SVs, age, institution, and menstrual cycle phase were included in models 4 and 5. Specifically, the variance explained by DNAm increased to 18.25%, and the variance for common genetic variants increased to 23.78% when all covariates were included. Moreover, we observed a significant reduction in variance explained by DNAm from model 1 (19.58%) to model 3 (12.35%) when the GRM was included in the model. This suggests that genetic regulation of methylation and population structure, as captured by the GRM; age; institution; and technical variation (SVs) contributed to the variation in methylation in endometrium and, therefore, influenced the relationship between DNAm and endometriosis status and should be accounted for in the subsequent analyses. The last model that includes all potential covariates (ORM + GRM + SVs + age + institution + menstrual cycle phase) was then used to develop the MRS.

### 2.3. MRS Captures DNAm Differences Between Endometriosis Cases and Controls

An outline of the MRS development and evaluation pipeline is illustrated in Figure 1. We estimated the effect sizes of DNAm probes on endometriosis via MLM-based omic association (MOA), multi-component MLM-based association excluding the target (MOMENT), and best linear unbiased prediction (BLUP) using four different training sets, each excluding one of the four institutions (1—Centre for Inflammation Research, University of Edinburgh (CIR); 2—University of California San Francisco (UCSF); 3—Oxford Endometriosis CaRe Centre (Oxford); 4—Institute for Molecular Bioscience (IMB)). This leave-one-institute-out approach ensures that the associations observed are not driven by a single institute and account for institute-specific sources of biases. Covariates included age, institution, menstrual cycle phase, and genetic PCs. Manhattan plots were plotted to show the statistical significance (*p*-value) of the associations between each DNAm probe (*n* = 762,651) and endometriosis, generated using the MOA (Appendix A) and MOMENT (Appendix A) methods. As shown in Appendix A, probe cg04415176, located at chromosome 2 and annotated to the *HOXD13* gene, was shown to be significantly associated with endometriosis according to the Bonferroni-corrected threshold of *p* < 6.56 × 10^−8^ when MOA and MOMENT were performed on Training Set 1. However, no probes were found to be significantly associated with endometriosis when MOA and MOMENT were performed with the other training sets.

To assess the consistency and reliability of the MRS generated, we calculated the correlation of effect sizes generated between all three methods, and the results showed that the correlations between MOA and MOMENT, MOA and BLUP, and MOMENT and BLUP were all >0.92 (Appendix A). Summary statistics, including effect sizes and *p*-values, of all DNAm probes used for generating the MRSs across all three methods on each training set are presented in Appendix A. We also computed the correlation between the MRSs applied to each test set across different MRS models illustrated in Appendix A. Each MRS model differs in terms of the effect size and DNAm probes selected for calculating the score. For effect sizes calculated via MOA and MOMENT, various *p*-value thresholds representing the significance of the relationship between the DNAm probe and endometriosis were used for probe selection. Alternatively, all DNAm probes were included in the score calculation when probe effect sizes were estimated via BLUP. Across all test sets, the correlation between all MRS models was positive, with a moderate-to-strong correlation (r = 0.59–0.99) between all models except for models with a *p*-value threshold of 1 × 10^−4^ and 1 × 10^−5^. This could potentially be driven by the lower number of DNAm probes used to generate MRS for both thresholds and, therefore, less reliable and unstable estimates.

To evaluate the performance of MRS generated across different models on the classification of endometriosis case–control status, we calculated the area under the receiver-operator curve (AUC) of each MRS model across different test sets. Three classification models were used to calculate the AUC: (1) case–control status was classified by MRS alone (AUC_1_), (2) case–control status was classified by both MRS and polygenic risk score (PRS) (AUC_2_), and (3) case–control status was classified by PRS only (AUC_3_). AUCs for all MRS models across all test sets were included in Appendix A. Overall, the choice of MRS models in general did not result in a statistically significant difference in prediction accuracy. Yet, to provide a more thorough and consistent comparison between test sets, the following selection criteria were used to select the appropriate MRS models that generated the maximum performance within each test set:MRSs derived from *p*-value thresholds of 1 × 10^−4^ and 1 × 10^−5^ were excluded.MRSs that yielded the highest AUC within the test set and classification model and demonstrated a significant association with endometriosis were selected.If none of the MRSs had a significant association, the MRSs with the highest AUC were chosen.

For the MRS-only classification model, MRSs generated on the CIR test set using MOA and a *p*-value threshold of 0.001 (AUC_1_ = 0.6748 [CI = 0.5468–0.8029]) performed the best. The maximum AUC of all test sets was higher than 0.5, ranging from 0.5555 to 0.6748. MRSs generated in three of the four test sets showed a significant association with endometriosis (*p* < 0.05). MRS models that generated the maximum performance also vary across test sets, involving both MOA and MOMENT methods and a range of *p*-value thresholds from 0.001 to 0.2. (Figure 2a).

### 2.4. Unique Contribution of MRS in the Case–Control Classification of Endometriosis

The performance of a combined-risk-score classification model was evaluated to demonstrate whether MRS can contribute any additional classification value to the current PRS. Figure 2 and Figure 3 show that, in most instances, the MRS + PRS classification model has higher accuracy when compared to the MRS-only classification model. CIR had the highest combined model AUC_2_ performance of 0.7284 [CI = 0.6075–0.8493]. Statistical significance between MRS and endometriosis in the combined model was only observed in the CIR (*p* = 0.0072) test set, yet marginal significance was also seen in the UCSF (*p* = 0.0541) and IMB (*p* = 0.0697) test sets. To further validate the unique contribution of MRS, independent of PRS, in the classification of endometriosis case–control status, the performance of PRS is calculated and compared with the MRS classification models. The performance of the MRS + PRS (AUC_2_ range from 0.5539 [CI = 0.4439–0.6640] to 0.7284 [CI = 0.6075–0.8493]) classification model was consistently higher than the PRS-only model (AUC_3_ range from 0.5123 [CI = 0.4011–0.6235] to 0.6342 [CI = 0.5083–0.7601]) (Figure 2 and Figure 3). Overall, we observed that MRS offers a distinct contribution to the case–control classification of endometriosis beyond what is provided by the PRS.

### 2.5. Sensitivity Analyses: Performance of MRS Within European-Ancestry Samples

Weightings used to compute the PRS in this study were developed from European-ancestry cohorts. Therefore, the performance of PRS may be underestimated when applied to a multi-ancestry cohort. Hence, we aimed to verify our results by restricting the development and evaluation of MRS analyses to only participants of European genetic ancestry. Appendix A shows the distribution of participants’ genetic ancestry across institutions. After removing non-European-ancestry participants, a total of 79, 215, 126, and 188 samples from CIR, IMB, Oxford, and UCSF, respectively, that had both DNAm and genotyping data were used for subsequent analyses.

Overall, consistent with the results shown in the multi-ancestry samples, the maximum AUC for the MRS + PRS classification model was higher than the AUC for PRS across all test sets, as shown in Figure 4. The maximum AUC of the MRS + PRS classification model across all test sets ranges from 0.5148 (CI = 0.4014–0.6282) to 0.6952 (CI = 0.5760–0.8144), while the maximum AUC of the PRS-only classification model ranges from 0.4870 (CI = 0.3706–0.6035) to 0.6366 (CI = 0.5090–0.7641). None of the MRSs developed exhibit statistically significant association with endometriosis. Details about DNAm probes used for developing the MRSs, including the effect size estimated from MOA, MOMENT, and BLUP across all test sets, were included in Appendix A. AUCs of each MRS model across all test sets were also reported in Appendix A.

## 3. Discussion

Studies have explored the association between common germline genetic variants and endometriosis, providing substantial evidence for the genetic contribution to the disease’s aetiology. Still, research on the environmental impact on endometriosis has been limited [5]. Both genetic effects [9,10] and effects of environmental exposures can contribute to variation in DNAm [38,39,40,41]. This makes it a valuable biological mediator for the effects of environmental exposures on complex diseases. However, previous methylation studies in the endometrium could not pinpoint and replicate individual DNAm sites associated with the disease due to the limited sample size and small effect sizes. This study identifies associations between effects aggregated across several DNAm sites, in the form of an MRS, and disease risk and highlights contributions of methylation signals to endometriosis independent of common genetic variants.

Using endometrial methylation data from 881 women, we estimated that 18.25% of the variance in endometriosis case–control status is captured by DNAm. Notably, this estimate was computed after correcting for common genetic variants, which were estimated to capture 23.78% of endometriosis variance, emphasising that DNAm signals associated with endometriosis could be derived from non-genetic factors, i.e., environmental factors. However, the results do not exclude the possibility that somatic mutations, rare variants, and structural variants may also influence DNAm signals associated with endometriosis. Various studies have estimated the amount of phenotypic variance explained by methylation, and values can range from 2.88% to 61.14%, depending on the nature of the trait. For example, body fat and adiposity-related biochemical traits, including body fat percentage (61.14%) [42] and glucose level (29.07%) [42], known to be driven by a large component of nongenetic factors, are reporting higher variance explained by DNAm compared to complex disease traits like endometriosis, Parkinson’s disease (21%) [43], and autism spectrum disorder (2.88%) [44]. Notably, the estimates for Parkinson’s disease and autism spectrum disorder do not account for genetic effects. Moreover, the contribution of common genetic variants to endometriosis, without accounting for DNAm, estimated in this study (28.83%) was also similar to previously published estimates of 26% [45]. Previous studies examining the interplay between DNAm and common genetic variants on disease variance have shown additive effects by estimating and comparing variance explained by PRS and MRS individually and combined. For instance, in major depressive disorder, the variance explained by both PRS and MRS combined (3.99%) was additive to the variance explained by PRS (2.40%) and MRS (1.75%) alone [46]. Similarly, we observed that the variance in endometriosis captured by both DNAm and common genetic variants combined (ORM (12.35%) + GRM (22.38%) = 34.73%) was higher than DNAm (19.58%) and common genetic variants (28.83%) alone. This suggests an additive contribution of each factor to endometriosis. However, it is worth noting that the variances explained and observed in this study were not truly additive. Some variances explained by DNAm overlapped with common genetic variants. For example, a recent discovery of 51 cis-mQTLs associated with endometriosis revealed genetic variants that can influence DNAm levels, highlighting the interplay between genetic factors and DNAm contributing to disease risk [26].

Aggregating the effects of multiple DNAm sites across the genome, we generated an endometriosis MRS using different computational approaches. The best-performing MRS with the highest AUC (0.68) was computed using MOA and a total of 746 DNAm probes (*p*-value threshold < 0.001), using CIR as the test set and the largest training set containing IMB, Oxford, and UCSF. This performance was comparable to other complex diseases, where previous studies have shown that MRSs developed for the prediction of breast cancer and amyotrophic lateral sclerosis had similar AUCs of 0.63 and 0.65, respectively [47,48]. We observed that the majority of MRSs with the highest AUC for each test set were generated using either MOA or MOMENT and not BLUP, suggesting that effect sizes estimated from MOA and MOMENT, combined with the targeted selection of DNAm probes, align more closely with the epigenetic architecture of endometriosis. This implies that the effects associated with endometriosis may be more concentrated within a specific subset of selected epigenetic markers [49]. Although pathway analysis could potentially provide more biological insights, it was not considered in this study as the number of probes used to calculate the MRSs is large, leading to an increased likelihood of identifying pathways that may not be relevant to endometriosis. Overall, the ability of MRS developed in this study to classify endometriosis cases and controls based on DNAm in the endometrium provides further evidence for DNAm differences between endometriosis cases and controls.

As reported previously, genetic risk variants in the form of PRS capture an increased risk of endometriosis [50]. To demonstrate the additional risk information captured by MRS independent of genetic risk, we calculated the PRS for participants in each test set. We compared its classification performance against the MRS + PRS model. Since the weightings of PRS were derived from predominantly European-ancestry datasets, it was expected to underperform among mixed-ancestry samples. Hence, results were further validated by restraining the risk score development and evaluation process to within European-ancestry samples only. A higher AUC when both MRS and PRS were included in the classification model compared to the PRS-only model was consistently observed across test sets and when models were restricted to include only European-ancestry participants. Similar trends have been observed in previous studies comparing the prediction accuracy of an MRS + PRS model with the single score model in BMI. The study showed that the MRS + PRS model had a larger prediction accuracy than the PRS-only model, suggesting that both scores acted additively and MRS could capture variance in BMI independent of the genetic determinants [51].

A strength of this study is the ability to train and test the performance of endometriosis MRS across multiple independent cohorts. However, variability in sample sizes, potential environmental exposures, cell composition, and independent processing of DNA samples between institutions may have influenced the results. While SVA analysis accounted for unknown variations, training and test sets were processed separately. Limited power, small effect sizes, and lack of significant *p*-values for individual DNAm sites and for some MRS models highlight the need for larger sample sizes and additional validation cohorts to verify the results. Additionally, although there is evidence of methylation differences between stage III/IV and controls, we did not consider developing MRS for the severe group only due to their much smaller sample size, which would further reduce power during evaluation.

Several observational studies have reported an association between environmental exposures and endometriosis [52,53,54,55]. DNAm serves as a valuable biomarker for environmental exposures [56], providing insights into how pollutants, diet, and other external factors contribute to disease pathogenesis. Identifying these influences could enhance our understanding of how epigenetic modifications regulate key biological mechanisms, including hormonal regulation, cellular proliferation, and inflammation. This knowledge may lead to the discovery of novel therapeutic targets, ultimately improving endometriosis management.

Epigenetic markers also show promise as biomarkers for disease detection and risk stratification alongside non-invasive approaches such as detecting microRNAs in liquid biopsy [57], autoantibodies in blood [58], or menstrual fluid analysis [59] and could facilitate early diagnosis and personalised risk assessment, enabling identification of high-risk individuals and informing targeted preventative and treatment strategies. However, further studies are needed to assess the utility of DNAm biomarkers for early diagnosis, given several known challenges such as tissue-specific DNAm patterns [60], the sensitivity of DNAm signals to external confounding environmental as well as technical factors, and the dynamic change in DNAm signals across the lifespan [61]. These issues will need to be addressed before DNAm biomarkers can be reliably implemented in clinical practice.

This study supports the hypothesis that DNAm influences endometriosis risk independently of genetic variation, emphasising the importance of molecular techniques in studying non-genetic factors. Integrating MRS with PRS has demonstrated an improved classification performance, reinforcing the predictive utility of epigenetic factors beyond common genetic variation. These findings underscore the need for comprehensive epigenetic studies to explore how environmental exposures contribute to endometriosis pathogenesis, paving the way for novel preventive and therapeutic approaches tailored to individual risk profiles.

## 4. Materials and Methods

### 4.1. Sample Collection and Processing

Data used for analyses in this study were generated from samples recruited as part of a previously published study [26]. Briefly, endometrial tissue samples were collected through case–control studies at four different institutions namely, the University of California San Francisco, California (480 samples); the University of Melbourne, Melbourne, Australia (315 samples); Oxford Endometriosis CaRe Centre, Oxford, UK (193 samples); and the EXPPECT Centre, The University of Edinburgh, Edinburgh, Scotland, UK (86 samples) and processed as described previously by Mortlock et al. [26]. The recruitment sample size was determined based on previous power calculations by Rahmioglu et al. [62], who estimated that to detect a 2% difference in 78% of the DNAm probes between cases and controls in the endometrium, a sample size of 500 is needed. Participants who had been on contraceptive steroids or gonadotropin-releasing hormone analogues during the 3 months prior to sampling, had undefined menstrual cycles, or had signs of endometrial hyperplasia or cancer were excluded from the recruitment process. A total of 679 surgically diagnosed endometriosis cases, 389 controls, and 6 individuals with unknown endometriosis status were recruited. Cases were defined as women who were surgically diagnosed with endometriosis, while controls comprised women without any visualised endometriosis during surgery and who had no history of endometriosis. To avoid bias, controls from all four institutions were recruited in approximately equal proportions to cases. DNAm measurements were calculated using Illumina Infinium MethylationEPIC Beadchips (Illumina, San Diego, CA, USA). Details of sample processing and preliminary QC are available in Mortlock et al. [26].

### 4.2. DNAm Quality Control

DNAm quality control and processing were performed as outlined in Nabais et al. [37] using the meffil R package. Low-quality samples and DNAm sites were excluded based on predetermined QC threshold parameters [37]. Technical variation was eliminated through functional normalisation, achieved by fitting linear models to probe intensity quantiles against control probe matrix principal components. After normalisation, the most variable probes were analysed for batch effects, regressing against factors like chip, chip column, and chip row. The significance threshold for association detection *p*-values is 0.01. Probes linked to sex chromosomes, those overlapping with SNPs, and those with non-unique hybridisation were removed based on recommended masking guidelines reported elsewhere [63]. Participants with unknown endometriosis status were removed. The final DNAm dataset consisted of a total of 318 controls and 590 cases with 762,651 DNAm sites.

### 4.3. Genotyping Data Quality Control

Samples passing the initial methylation QC were genotyped using Axiom Precision Medicine Research Array (Thermo Fisher Scientific, Waltham, MA, USA). Quality-controlled genotyping data and genetic ancestry used for analysis were obtained from Mortlock et al. 2023 [26]. Quality control was carried out separately for batch I and II samples. Steps include filtering out individuals with genotype call rates < 95%, a heterozygosity rate > 3 standard deviations away from the mean heterozygosity rate, and high relatedness (IBD > 0.2), as well as filtering variants with minor allele frequency < 5%, call rates < 95%, and deviation from Hardy–Weinberg equilibrium (*p*-value < 1 × 10^−5^). The data were pre-phased with SHAPEIT2 and imputed using the 1000 Genomes reference. The final genotype dataset consisted of 953 individuals (614 cases and 339 controls) and 5,201,970 common genetic variants. The genetic ancestry of all participants with genotyping information was identified using the 1000 Genomes P3v5 reference data and principal component analysis (PCA), as described in Mortlock et al. 2023 [26]. The total number of samples assigned for each ancestry is shown in Table 1.

### 4.4. Covariate Selection

Several studies have shown that age [64], ancestry [65], menstrual cycle [26], cell type proportion [11], and batch [66] play a role in the variation of DNAm profiles between individuals. Hence, these factors should be accounted for during analyses to remove any unwanted variation between the samples not contributed by the variables of interest, in this case, endometriosis status. Potential covariates selected for covariate selection analysis include age, menstrual cycle phase, institution, genetic ancestry, chip, sentrix ID, and batch. The definitions for each potential covariate were as follows: Age: Participant’s self-reported chronological age treated as a continuous variable. Menstrual cycle phase: In short, menstrual cycle phase was assigned to specimens as a categorical variable in several phases based on the criteria of Noyes et al. [67] as described in Mortlock et al. 2023 [26]: menstrual, early proliferative (EP), mid-proliferative (MP), late proliferative (LP), early secretory (ESE), mid-secretory (MSE), and late secretory (LSE). All proliferative samples were grouped as PE and unassigned secretory sub-phase samples as SE. Institutions: A categorical variable referring to the sites where tissue samples were analysed. Abbreviations were included in brackets. Samples obtained from the EXPPECT Centre were analysed at the Centre for Inflammation Research, University of Edinburgh (CIR); samples collected by the University of Melbourne were analysed at the Institute for Molecular Bioscience, University of Queensland (IMB); samples collected from the University of California San Francisco were both collected and analysed at the University of California San Francisco (UCSF); and samples from the Oxford Endometriosis CaRe Centre were analysed at the same centre (Oxford). Genetic ancestry: A categorical variable that represents participants’ ancestry assigned using their genotype data.

Statistical tests, including the *t*-test, chi-squared test, and Fisher’s exact test, were conducted to identify potential covariates that are significantly associated with endometriosis status. The t-test was applied to continuous covariates, while the chi-squared test was used for categorical variables. When the assumptions of the chi-squared test were not met, Fisher’s exact test was used instead.

PCs of DNAm beta values were generated using the Omic-data-based Complex Trait Analysis (OSCA) software, and PCA plots were analysed to identify any variation in sample clustering based on covariates. Statistical tests, including ANOVA for categorical covariates and linear regression for continuous covariates, were performed to identify any significant association between the top 15 DNAm PCs and potential covariates.

Potential covariates that were either significantly associated with endometriosis status (*p*-value < 0.05) or DNAm PCs (*p*-value < 0.05 or showed differential clustering on PCA plots) were included as continuous covariates in the downstream analyses. All statistical tests were performed using R version 4.3.2.

### 4.5. Surrogate Variable Analysis

SV analyses were applied to eliminate batch effects and any hidden sources of variation not addressed by the selected covariates. Analysis was conducted using the R package SmartSVA. Briefly, the residuals of the linear model, where DNAm m-values were modelled as a function of endometriosis case–control status, were computed. A full model matrix, derived from the endometriosis case–control status of the samples, and a null model matrix were generated. SV is a continuous variable that represents sources of variation in DNAm values not contributed by endometriosis status and were estimated by combining all three components together. All SVs estimated were used to adjust the DNAm values prior to analysis.

SVs were generated from all the DNAm samples in this study (908 samples) and applied to the estimation of the proportion of variance in endometriosis captured by DNAm. Similarly, SVs were generated in training and test samples separately during the development of MRSs.

### 4.6. Estimation of the Proportion of Variance in Endometriosis Captured by DNAm

The proportion of variance in endometriosis risk that can be captured by common genetic variants (SNPs) alone was estimated using genome-based restricted maximum likelihood (GREML) from the GCTA software (version 1.94.1). Similarly, the proportion of variance in endometriosis risk between cases and controls that can be captured by DNAm was estimated using OREML from the OSCA (version 0.46) software [49]. Briefly, an ORM was generated from DNAm beta values of endometrium samples. The proportion of the trait variance captured was then estimated from the ORM using the OREML model. To calculate the proportion of variance in endometriosis captured by DNAm and common genetic variants combined, the ORM and GRM were generated from DNAm beta values and genotype data, respectively, and both were included in the OREML model simultaneously. The five OREML models used to estimate the proportion of variance in endometriosis captured were as follows:ORM;GRM;ORM + GRM;ORM + GRM + SV;ORM + GRM + SV + age + institution + menstrual cycle phase;
and the results were compared. In the first model, ORM was generated using DNAm beta values from 881 individuals and 762,651 DNAm sites. GRM was generated using 5,201,970 SNPs in samples that passed both DNAm and genotyping quality control (881 samples). In the third model, both ORM and GRM were included simultaneously in the model. In the fourth model, the ORM generated in the second model was adjusted using SVs mentioned above prior to the OREML analysis. Lastly, covariates, namely age, institution, and menstrual cycle phase, were included in the fifth model.

### 4.7. Genetic PC Computation

We calculated the genetic PCs from the genotype data of participants that passed both the DNAm and genotyping quality control (881 samples) to correct for population stratification between samples from different institutions and for genetic ancestry in subsequent analyses. Genetic PCs were computed using the --pca feature from the software GCTA (version 1.94.1) [68], and a total of 11 PCs were selected to include as a covariate in the later analyses.

### 4.8. Methylation Risk Score (MRS) Development

A summary of MRS development and evaluation is illustrated in Figure 1. To evaluate the performance of the MRS developed, all samples that passed both DNAm and genotyping quality control were separated into training and test sets according to institution, in which samples from each institution were iteratively selected as the test set, while the remaining samples were used as the training set prior to the calculation of the MRS. The purpose of the training set was to estimate the effects used to calculate the MRS, and these effects were then applied to the test set to evaluate the performance of the MRS developed in an independent sample.

A total of four different combinations of training and test sets were analysed using this approach, as shown in Figure 1. Training Set 1 denotes IMB, Oxford, UCSF as training (800 samples); Training Set 2 denotes IMB, Oxford, CIR as training (513 samples); Training Set 3 denotes IMB, UCSF, CIR as training (741 samples); Training Set 4 denotes Oxford, UCSF, CIR as training (589 samples). Training Sets 1, 2, 3, and 4 had CIR (81 samples), UCSF (368 samples), IMB (292 samples), and Oxford (140 samples) as test sets, respectively. SVA analyses were then performed on the training and test sets separately.

Estimation of the effect sizes for each DNAm probe was performed using three different methods: MOA, MOMENT, and BLUP from the OSCA software [49]. MOA and MOMENT are reference-free mixed-linear models and are used for identifying DNAm sites that are associated with a complex trait.

The equation of the MOA (1) and MOMENT (2) models were as follows:(1)$$y={\;w}_{i}b_{i}+C\beta +\mathrm{Wu}+e,$$(2)$$y={\;w}_{i}b_{i}+C\beta +\underset{j}{\sum }W_{j}u_{j}+e,$$

Briefly, y represents the phenotype values, w_i_ is the standardised DNAm values of the target probe i, b_i_ is the effect of probe i on the phenotype, C is a matrix for covariates, β is the effects of covariates on the phenotype, W is a matrix of DNAm values of all probes, u denotes the joint effects of all probes on the phenotype, e represents residuals, W_j_ is a matrix of DNAm values of the probes in the jth group (all probes except probes that are highly correlated to the target probe), and u_j_ represents the joint effects of probes in W_j_ on the phenotype. A summation term for W_j_ and u_j_ denotes that there could be more than one jth group of probes in the model, depending on the probe effect distribution. In summary, the difference between MOA and MOMENT is the way the effect sizes of each DNAm probe are generated. MOMENT first segregates probes into groups based on their probe effect distribution using linear regression and fits the two groups separately into the model, while MOA assumes all probe effects have a similar distribution and fits them as a single term into the model. In order to reduce convergence problems caused by too much variation explained by the first group of probes, a stepwise selection procedure was implemented in MOMENT to reduce the number of probes in the first group. Moreover, in MOMENT, probes that were highly correlated to the target probe, defined by probes that were located within 50 Kb in distance, were removed from the random-effect term of the model. MOMENT has been shown to be more powerful in correcting for potential confounders but with a slight loss in power compared to MOA [49]. This could reflect either the lack of power to detect true positives for MOMENT or potential false positives for MOA [47]. Hence, both models were used in this study, and the results were compared. Effect sizes for each DNAm probe were estimated by running --moa (MOA) and --moment2 (MOMENT) from the OSCA software on the selected training samples with endometriosis status of the samples as the phenotype. Covariates included in this analysis were age, institution, menstrual cycle phase, and genetic PCs. In addition to effect size, *p* values representing the statistical significance of associations between each probe and endometriosis were generated as well and were used as a criterion for probe selection. DNAm sites were mapped to the latest GRCh38 genome build and annotated to genes using GENCODE v41 to identify potential genes or regulatory regions associated with the site.

We also applied a genome-wide approach to estimate effect sizes, BLUP. BLUP is a statistical model that estimates the random effects in a mixed-linear model from the variance–covariance matrices of the random effects, the phenotype data, and the fixed effect terms [69]. In this study, effect sizes for each DNAm probe were derived from the overall effect of all DNAm probes, as represented by the ORM of the training individuals. The ORM was first generated from DNAm beta values of the training samples using OSCA. The joint effect of all probes on the phenotype for each sample in the training set was then estimated using --reml-pred-rand in OSCA with ORM, endometriosis status, and covariates including age, institution, menstrual cycle phase, and genetic PCs incorporated as input. Effect sizes of each probe were then predicted from the joint effect of all probes using the --blup-probe feature. Since BLUP probe effects were estimated from the aggregated effects of all probes, all DNAm probes (*n* = 762,651) were included to generate the MRS.

The definition of MRS is the addition of an individual’s weighted methylation markers’ beta values of a set of CpG sites as indicated in the formula:MRS_i_ = w_1_m_i1_ + … + w_k_m_ik_(3)
where w stands for the weights or effect size of each of the DNAm sites or markers, m is the methylation beta values, and k is the number of pre-selected methylation probes [35]. The score represents the collective effect of several DNAm sites of an individual. MRS was calculated on the test samples using their DNAm beta values, and the effect size of selected probes was estimated from the three models mentioned above according to Equation (3) using R version 4.2.1.

### 4.9. Correlation Between Effect Sizes Generated from MOA, MOMENT, and BLUP

To provide a more comprehensive assessment of the MRS, the correlation between effect sizes generated from the three different methods, MOA, MOMENT, and BLUP, was determined. For each training set, a pairwise comparison of effect sizes generated from all methods was performed. Pearson correlation coefficients were calculated using the cor() function in R version 4.3.2, and correlation plots were used to visualise the relationship.

### 4.10. MRS Evaluation

To assess the accuracy of these profile scores in classifying samples into endometriosis cases and controls, the AUC was employed. This curve illustrates the relationship between the false positive rate (specificity) and the true positive rate (sensitivity) in logistic regression. MRS was evaluated using case–control status as the outcome variable and MRS as the predictor of the logistic regression. The R version 4.2.1 package pROC was utilised for generating receiver-operator characteristic curves and calculating the AUC for each profile score [70]. The 95% confidence intervals for the AUC were calculated using the ci.auc function, employing the DeLong method.

### 4.11. Correlation Between MRSs Generated from Different MRS Models

Pairwise correlation analysis was carried out on MRS calculated from different models for each test set using the cor() function in R version 4.2.1. Correlation matrix and coefficients were plotted to visualise the relationship between MRSs derived from different models.

### 4.12. PRS Development and Evaluation

To compare the utility of MRS in the classification of endometriosis case–control with and without PRS, we calculated the PRS of participants in each test set and compared its performance in classifying endometriosis with the other two corresponding MRS classification models using the following models: (1) case–control status as the outcome variable and PRS as the predictor, and (2) case–control status as the outcome variable and MRS and PRS as the predictor of the logistic regression model. PRS was computed using the participant’s genotyping data as input and the plink2 score function on weightings generated from McGrath et al. [6]. Similar to evaluating the performance of MRS, the performance of PRS in classifying endometriosis case–control was identified by calculating the AUC of the logistic regression model where endometriosis case–control status was used as an outcome and PRS as the predictor. Notably, to provide a rigorous comparison between classification models, the AUC of the PRS logistic regression model was generated separately on each test set, and comparisons were made within test sets.

## Figures and Tables

**Figure 1 ijms-26-03760-f001:**
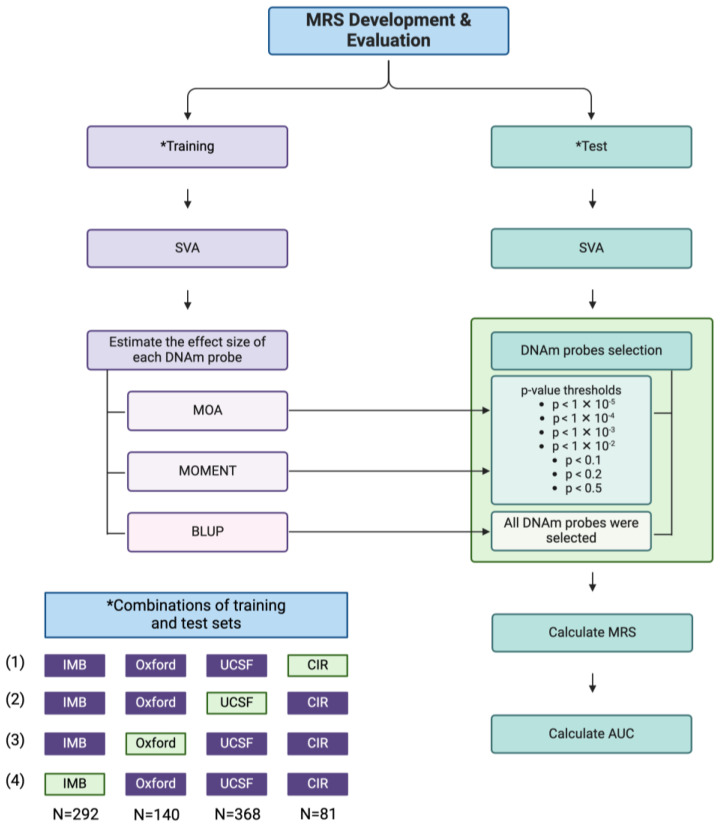
Methylation risk score (MRS) development and evaluation pipeline. Outline of steps used to generate MRS. Samples (N = 881) were split into training and test sets according to the institutions, as shown in the bottom left panel. Four different training and test set combinations were formed (purple denoted training set; green represented test set), and the same MRS development and evaluation process was applied to each combination. The asterisk indicates the parts of the flowchart where different combinations of training and test sets are applied. The effect size of DNAm probes was estimated using MLM-based omic association (MOA), multi-component MLM-based association excluding the target (MOMENT), and best linear unbiased prediction (BLUP). DNAm probes to be included in the MRS were selected according to their *p*-value threshold for MOA and MOMENT, while all DNAm probes were included in the MRS for BLUP. MRS was calculated on the test samples according to the features selected from the training set (effect size and DNAm probes), and the area under the receiver-operator curve (AUC) was computed to evaluate the performance of the MRS. Figure created in BioRender, https://BioRender.com/n39u049 (accessed on 28 January 2025).

**Figure 2 ijms-26-03760-f002:**
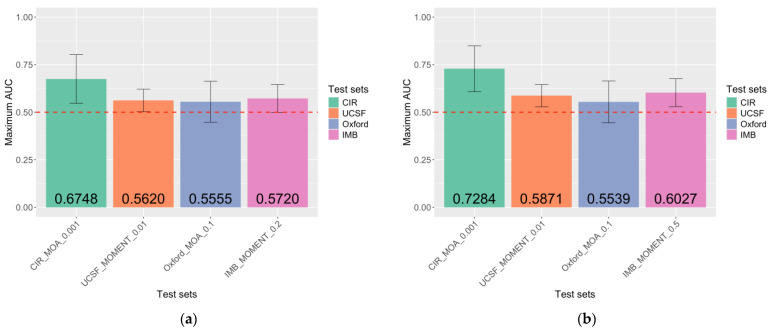
Maximum accuracy across different endometriosis MRS methods for each test set. (**a**) The accuracy of the MRS-only classification model, in which case–control status was the outcome variable, and MRS was the predictor. CIR_MOA_0.001: *n* = 746, *pMRS* = 0.0073; UCSF_MOMENT_0.01: *n* = 7447, *pMRS* = 0.0440; Oxford_MOA_0.1: *n* = 76,350, *pMRS* = 0.2695; IMB_MOMENT_0.2: *n* = 152,686, *pMRS* = 0.0494. (**b**) Accuracy for the MRS + polygenic risk score (PRS) classification model, in which case–control status was the outcome variable and MRS and PRS were the predictors. CIR_MOA_0.001: *n =* 746, *pMRS* = 0.0072, *pPRS* = 2.15 × 10^−2^; UCSF_MOMENT_0.01: *n* = 7447, *pMRS* = 0.0541, *pPRS* = 3.41 × 10^−2^; Oxford_MOA_0.1: *n* = 76,350, *pMRS* = 0.2643, *pPRS* = 4.30 × 10^−1^; IMB_MOMENT_0.5: *n* = 382,410, *pMRS* = 0.0697, *pPRS* = 2.15 × 10^−2^. The X-axis shows the test set names, the method for estimating DNAm probe effect sizes, and the *p*-value threshold for probe selection. “*n*” refers to the number of DNAm probes. The Y-axis plots AUCs, with error bars representing 95% confidence intervals. pMRS and pPRS represent the *p*-values from logistic regression for the association between endometriosis and MRS or PRS, respectively.

**Figure 3 ijms-26-03760-f003:**
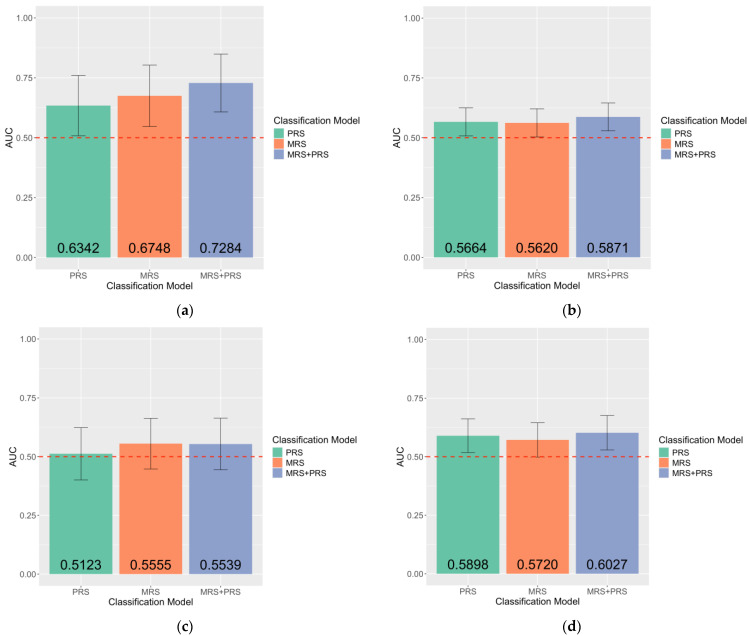
Accuracy of endometriosis case–control classification increased with the inclusion of both MRS and PRS: (**a**) CIR test set (PRS: *n* = 784,516, *pPRS* = 2.04 × 10^−2^; MRS: *n* = 746, *pMRS* = 0.0073; MRS + PRS: *n* = 746, *pMRS* = 0.0072, *pPRS* = 2.15 × 10^−2^); (**b**) UCSF test set (PRS: *n* = 784,516, *pPRS* = 0.0279; MRS: *n* = 7447, *pMRS* = 0.0440; MRS + PRS: *n* = 7447, *pMRS* = 0.0541, *pPRS* = 0.0341); (**c**) Oxford test set (PRS: *n* = 784,516, *pPRS* = 4.4 × 10^−1^; MRS: *n* = 76,350, *pMRS* = 0.2695; MRS + PRS: *n* = 76,350, *pMRS* = 0.2643, *pPRS* = 4.3 × 10^−1^); (**d**) IMB test set (PRS: *n* = 784,516, *pPRS* = 1.55 × 10^−2^; MRS: *n* = 152,686, *pMRS* = 0.0494; MRS + PRS: *n* = 382,410, *pMRS* = 0.0697, *pPRS* = 2.15 × 10^−2^). The number of DNAm probes included is denoted by *n*. Classification models were labelled on the X-axis. AUCs were plotted on the Y-axis and labelled at the bottom of each bar graph. Error bars indicate 95% confidence intervals. Red dashed lines indicate an AUC of 0.5. pMRS denotes *p*-values from the logistic regression model showing the association between endometriosis and MRS. Similarly, pPRS denotes *p*-values from the logistic regression model showing the association between endometriosis and PRS.

**Figure 4 ijms-26-03760-f004:**
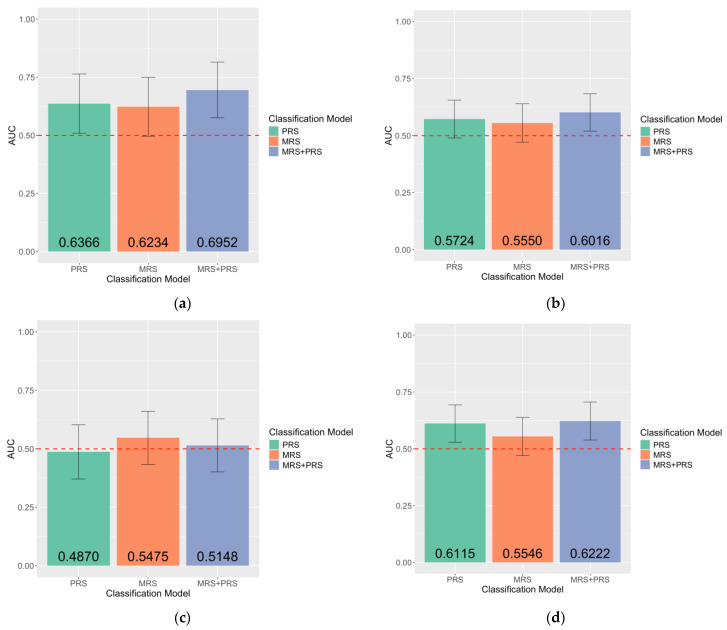
Accuracy of endometriosis case–control classification within European-ancestry samples increased with the inclusion of both MRS and PRS: (**a**) CIR test sets (PRS: *n* = 784,516, *pPRS* = 2.46 × 10^−2^; MRS: *n* = 381,992, *pMRS* = 0.0605; MRS + PRS: *n* = 762,651, *pMRS* = 0.0953, *pPRS* = 2.93 × 10^−2^); (**b**) UCSF test sets (PRS: *n* = 784,516, *pPRS* = 4.45 × 10^−2^; MRS: *n* = 7303, *pMRS* = 0.2367; MRS + PRS: *n* = 7303, *pMRS* = 0.1994, *pPRS* = 3.87 × 10^−2^); (**c**) Oxford test sets (PRS: *n* = 784,516, *pPRS* = 4.36 × 10^−1^; MRS: *n* = 752, *pMRS* = 0.4941; MRS + PRS: *n* = 762,651, *pMRS* = 0.5596, *pPRS* = 4.37 × 10^−1^); (**d**) IMB test sets (PRS: *n* = 784,516, *pPRS* = 1.65 × 10^−2^; MRS: *n* = 762,651, *pMRS* = 0.1009; MRS + PRS: *n* = 76,238, *pMRS* = 0.1808, *pPRS* = 2.32 × 10^−2^). The number of DNAm probes included is denoted by *n*. Classification models were labelled on the X-axis. AUCs were plotted on the Y-axis and labelled at the bottom of each bar graph. Error bars indicate 95% confidence intervals. Red dashed lines indicate an AUC of 0.5. pMRS denotes *p*-values from the logistic regression model showing the association between endometriosis and MRS. Similarly, pPRS denotes *p*-values from the logistic regression model showing the association between endometriosis and PRS.

**Table 1 ijms-26-03760-t001:** Sample information and differences between cases and controls.

Characteristics		Endometriosis Status		*p*-Value
		Controls (N = 318)	Cases (N = 590)	
Age Mean [95% CI] (range) *t*-test		37 [36.1–37.9] (18–55) (N = 314)	34.2 [33.6–34.8] (18–53) (N = 587)	1.29 × 10^−6^
Menstrual cycle phase N (%) Chi-squared	Proliferative	154 (48.4%)	285 (48.3%)	0.75
	Secretory (undefined sub-phase)	7 (2.2%)	14 (2.4%)	
	Early secretory	41 (12.9%)	71 (12.0%)	
	Mid-secretory	72 (22.6%)	121 (20.5%)	
	Late secretory	33 (10.4%)	66 (11.2%)	
	Menstrual	11 (3.5%)	33 (5.6%)	
Institutions N (%) Chi-squared	CIR ^1^	31 (9.7%)	52 (8.8%)	8.11 × 10^−5^
	IMB ^2^	83 (26.1%)	213 (36.1%)	
	Oxford ^3^	41 (12.9%)	110 (18.6%)	
	UCSF ^4^	163 (51.3%)	215 (36.4%)	
Genetic ancestry N (%) Chi-squared	ADMIX	24 (7.5%)	49 (8.3%)	1.89 × 10^−6^
	African	33 (10.4%)	13 (2.2%)	
	American	21 (6.6%)	29 (4.9%)	
	Eastern Asian	25 (7.9%)	47 (8.0%)	
	European	207 (65.1%)	417 (70.7%)	
	Southern Asian	8 (2.5%)	35 (5.9%)	

^1^ Centre for Inflammation Research, University of Edinburgh. ^2^ Institute for Molecular Bioscience, University of Queensland. ^3^ University of California San Francisco. ^4^ Oxford Endometriosis CaRe Centre.

**Table 2 ijms-26-03760-t002:** Proportion of variance in endometriosis status captured by DNA methylation (DNAm) and common genetic variants.

No.	OREML Models	Proportion of Variance Captured ^2^ (s.e. ^a^)			Phenotypic Variance ^1^ (s.e. ^a^)
		ORM ^b^	GRM ^c^	ORM + GRM ^e^	
1	ORM ^b^	19.58% (0.07)	-	-	0.2481 (0.02)
2	GRM ^c^	-	28.83% (0.17)	-	0.2251 (0.01)
3	ORM ^b^ + GRM ^c^	12.35% (0.06)	22.38% (0.15)	34.73%	0.2361 (0.01)
4	ORM ^b^ + GRM ^c^ + surrogate variable (SVs) ^d^	10.70% (0.07)	27.94% (0.16)	38.64%	0.2251 (0.01)
5	ORM ^b^ + GRM ^c^ + SVs ^d^ + age + institution + menstrual cycle phase	18.25% (0.08)	23.78% (0.15)	42.03%	0.2187 (0.01)

^1^ Phenotypic variance denotes the variability in endometriosis status among the sample population. It quantitatively measures the extent to which the two possible outcomes of endometriosis (cases and controls) vary within the sample population. ^2^ Proportions of variance captured were shown in percentages. They were calculated using ( [variance captured by the omics data/phenotypic variance]×100%). ^a^ s.e. represents standard error. ^b^ Omics relationship matrix (ORM) represents the omics relationship matrix derived from DNAm beta values from endometrium samples, which also denotes the proportion of endometriosis status variance captured by DNAm. ^c^ Genomic relationship matrix (GRM) represents the genomic relationship matrix derived from the genotype data of the samples, which also indicates the proportion of endometriosis status variance captured by common genetic variants. ^d^ DNAm values had been preadjusted for batch effects using SVA. ^e^ The proportion of endometriosis status variance captured by DNAm and common genetic variants is combined and computed by adding the captured variance by ORM + GRM within their respective models.

## Data Availability

All data used in this study were obtained from Mortlock et al. [26]. Methylation data have been deposited and are available from GEO (GEO: GSE223817). Genotype data generated are available upon approval from dbGAP (phs003307.v1). Code used to run the analyses is available on github: https://github.com/Li-Ying-Thong/Methylation-Risk-Score-DNA-Methylation-Endometriosis.git (accessed on 5 February 2025).

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
