# Peer review of "Methylation Risk Score Modelling in Endometriosis: Evidence for Non-Genetic DNA Methylation Effects in a Case–Control Study"

_ijms, 2025, doi:10.3390/ijms26083760_

Round 1

Reviewer 1 Report

Comments and Suggestions for Authors

Epigenetic processes are increasingly recognized as an important biological factor associated with endometriosis. The purpose of this study was to develop a methylation risk score using methylation data for endometriosis.  This work provides a valuable resource for understanding the impact of methylation on endometrial biology. This research points to DNAm as a potential biomarker of disease, highlighting the importance and need for more comprehensive epigenetic studies to identify signals associated with environmental exposures. 

However, the work is difficult to read. It may be of value to a narrow circle of specialists. Therefore, I suggest that the authors should emphasise in conclusion what exactly the objectives of the work have been achieved. 

Author Response

Comments 1: However, the work is difficult to read. It may be of value to a narrow circle of specialists. Therefore, I suggest that the authors should emphasise in conclusion what exactly the objectives of the work have been achieved.

Response 1:

Thank you for this suggestion to strengthen our conclusions. We have more clearly defined the aim and hypothesis in the introduction on page 3, lines 90 to 95. “We aimed to develop a MRS for endometriosis using endometrial methylation data from 1074 individuals to detect DNAm signals associated with endometriosis and investigate the unique non-genetic contribution of the DNAm to endometriosis risk. We hypothesise that DNAm contributes to endometriosis risk independently of genetic variation and that a MRS derived from endometrial methylation data can effectively identify DNAm signals associated with the disease.”  Similarly, we clarified the conclusion with regards to this aim and hypothesis on page 12, lines 396 – 400. “The study supports the hypothesis that DNAm influences endometriosis risk independently of genetic variation, highlighting the value of molecular techniques in studying non-genetic factors. The MRS effectively identified DNAm signals associated with the disease, and when combined with the PRS, it improved classification performance, reinforcing the predictive value of epigenetic factors beyond common genetic variation.”

Reviewer 2 Report

Comments and Suggestions for Authors

I read with great interest this Manuscript, which falls within the aim of the Journal.
Honestly, the topic is interesting enough to attract the readers’ attention. The methodology is accurate, and the data analysis supports conclusions. Nevertheless, authors should clarify some points and improve the discussion by citing relevant and novel critical articles about the topic:

-The title effectively conveys the study's focus, but it does not explicitly indicate the study design. Including "case-control study" or a similar descriptor in the title

-The background provides a solid rationale for investigating DNA methylation in endometriosis. However, it does not clearly specify a hypothesis.

-The study design is not explicitly mentioned early in the methods section. Clearly stating that this is a case-control study would enhance clarity.

-details on the recruitment period and data collection timeline are missing.

-The eligibility criteria for participants (cases and controls) are not fully detailed.

-The study does not mention any explicit efforts to address bias in participant selection, sample processing, or data interpretation. 

-The study should describe how the sample size was determined, including any power calculations used to ensure adequate detection of associations.

-The handling of missing data is unclear

-Results should not include consideration of similar articles but just evidence from this research. Please move it to discussion

-The categorization of continuous variables is not explained well. For example, age is included as a variable, but there is no clear justification for how it was handled in statistical models

-The key findings are well summarized, but their implications for clinical practice or future research are not deeply explored, like potential use for early diagnose with liquid biopsy (see doi: 10.3390/ijms24076116) or the mechanism of prevention with inflammation control (see doi: 10.3390/medicina59020347)

- Ethical approval and informed consent procedures should be more explicitly mentioned

Author Response

Comments 1: The title effectively conveys the study's focus, but it does not explicitly indicate the study design. Including "case-control study" or a similar descriptor in the title

Response 1: Thank you for this suggestion. We have now included reference to the study design in the title of the manuscript. “Methylation Risk Score Modelling in Endometriosis: Evidence for Non-Genetic DNA Methylation Effects in a Case-Control Study”

Comments 2: The background provides a solid rationale for investigating DNA methylation in endometriosis. However, it does not clearly specify a hypothesis.

Response 2: Thank you for pointing this out. We have added a clear hypothesis statement on page 3, lines 90 to 95: “We aimed to develop a MRS for endometriosis using endometrial methylation data from 1074 individuals to detect DNAm signals associated with endometriosis and investigate the unique non-genetic contribution of the DNAm to endometriosis risk. We hypothesise that DNAm contributes to endometriosis risk independently of genetic variation and that a MRS derived from endometrial methylation data can effectively identify DNAm signals associated with the disease.”

Comments 3: The study design is not explicitly mentioned early in the methods section. Clearly stating that this is a case-control study would enhance clarity.

Response 3: As suggested we have now highlighted the study design earlier in the methods on page 13, lines 415 to 416. “Briefly, endometrial tissue samples were collected through case-control studies at four different institutions namely, the University of California San Francisco, California (n = 480 samples); University of Melbourne, Melbourne, Australia (n = 315 samples); Oxford Endometriosis CaRe Centre, Oxford, UK (n = 193 samples); and the EXPPECT Centre, The University of Edinburgh, Edinburgh, Scotland, UK (n = 86 samples) and processed as described previously by Mortlock et al. [27].”

Comments 4, 5, 8, 12: details on the recruitment period and data collection timeline are missing; The eligibility criteria for participants (cases and controls) are not fully detailed; The handling of missing data is unclear; Ethical approval and informed consent procedures should be more explicitly mentioned

Response 4, 5, 8, 12: Thank you for highlighting this, and we appreciate the opportunity to clarify this further. In this study, we did not recruit any new participants or generate new data. Instead, we utilised previously collected data from Mortlock et al. 2023 [27]. As such, all available information regarding recruitment, data collection timelines, eligibility criteria, missing data management, ethical approval and informed consent is based on what was reported in the original study. To clarify this, we have added the following sentence in the methods section, page 13, line 414. “Data used for analyses in this study was generated from samples recruited as a part of a previously published study [27]. Briefly, endometrial tissue samples were collected through case-control studies at four different institutions namely,…….” Handling of missing phenotype data and poor-quality probes and samples is outlined in sections 4.2 and 4.3.

Comments 6: The study does not mention any explicit efforts to address bias in participant selection, sample processing, or data interpretation.

Response 6: Thank you for the feedback. The information regarding addressing bias in participant selection is provided on page 13, lines 430 to 431: “To avoid bias, controls from all four institutions were recruited in approximately equal proportions to cases.” Bias in sample processing has been accounted for statistically using surrogate variable analysis to eliminate batch effects and any hidden sources of variation not addressed by the selected covariates. The purpose of using surrogate variable analysis has been mentioned on page 14, lines 499 to 500. Additionally, covariates including age, institutions, menstrual cycle phase and genetic PCs were added into the model to develop MRS that is not biased against external confounders. [page 5, section 2.3 MRS captures DNAm differences between endometriosis cases and controls, lines 173 to 174, “Covariates included age, institutions, menstrual cycle phase and genetic PCs.”]. We also acknowledged that there may still be a bias in signals between different institutes. Henceforth, we have used a leave-one-out approach when developing and evaluating the MRS, using four variations of training sets, each excluding one institute.

Comments 7: The study should describe how the sample size was determined, including any power calculations used to ensure adequate detection of associations.

Response 7: Thank you for pointing this out. We agree that the rationale behind the sample size collected should be mentioned in the manuscript. Therefore, the following statement has been added on pages 13, lines 420 to 423. “The recruitment sample size was determined based on previous power calculations by Rahmioglu et al., who estimated that to detect a 2% difference in 78% of the DNAm probes between cases and controls in the endometrium, a sample size of 500 is needed.”

Comments 9: Results should not include consideration of similar articles but just evidence from this research. Please move it to discussion

Response 9: Thank you for highlighting this. We agree that results should not include comparing evidence to existing literature. Therefore, we have moved the following sentence “Several studies have shown that age [39], ancestry [40], menstrual cycle [27], cell type proportion [12] and batch [41] play a role in the variation of DNAm profiles between individuals. Hence, these factors should be accounted for during analyses to remove any unwanted variation between the samples not contributed by the variables of interest, in this case, endometriosis status.” from page 3, lines 100 to 104 (results) to page 14, lines 463 to 467 (methods). We have also moved the following sentence “As reported previously, genetic risk variants in the form of PRS capture an increased risk of endometriosis [55].” from page 8, line 252 (results) to page 12, lines 370 to 371 (discussion).

Comments 10: The categorization of continuous variables is not explained well. For example, age is included as a variable, but there is no clear justification for how it was handled in statistical models.

Response 10: Thank you for pointing this out. We agree that more clarification should be made on how the variables were handled statistically. Hence, we have made the following changes on page 14, lines 469 to 470, “The definitions for each potential covariate were as follows: Age: Participant’s self-reported chronological age and were treated as a continuous variable”. The type of variable of menstrual cycle phase (categorical), institutions (categorical), DNAm PCs (continuous) and surrogate variables (continuous) were also clarified in lines 471, 475, 497 and 505, respectively.

Comments 11: The key findings are well summarized, but their implications for clinical practice or future research are not deeply explored, like potential use for early diagnose with liquid biopsy (see doi: 10.3390/ijms24076116) or the mechanism of prevention with inflammation control (see doi: 10.3390/medicina59020347)

Response 11: Thank you for this suggestion. We have now summarised some potential clinical implications for the study and suggestions for future research into the concluding paragraph page 12, lines 396 to 411:

"The study supports the hypothesis that DNAm influences endometriosis risk independently of genetic variation, highlighting the value of molecular techniques in studying non-genetic factors. The MRS effectively identified DNAm signals associated with the disease, and when combined with the PRS, it improved classification performance, reinforcing the predictive value of epigenetic factors beyond common genetic variation. The findings underscore the need for more comprehensive epigenetic studies to explore how environmental exposures, such as pollutants and diet, contribute to endometriosis pathogenesis, potentially leading to novel preventive and therapeutic approaches [54]. DNAm shows promise as a biomarker for the disease, especially if detectable in accessible tissues like blood or menstrual fluid, enhancing early diagnosis and risk assessment. Such biomarkers could help identify high-risk individuals and guide targeted preventative measures [55]. Biomarker profiles may also inform personalised treatment approaches, optimising therapeutic response [56]. Additionally, understanding how epigenetic changes influence mechanisms like hormonal regulation, proliferation and inflammation could reveal new therapeutic targets that could help reduce symptoms, alleviate pain and improve endometriosis management [57]."

Round 2

Reviewer 2 Report

Comments and Suggestions for Authors

Thank you for having addressed our previous comments and for the improvements made to the manuscript. However, there are still a few issues that need to be resolved before publication:

  • The "Materials" section should be moved before the "Results", as it provides essential context needed to fully understand and interpret the findings.

  • Some of the citations (e.g., references 52, 55, 56, and 57) appear not to align well with the statements they are intended to support. We recommend reviewing these references to ensure consistency and relevance.

  • The author Mortlock is self-cited five times, which exceeds the recommended limit of three self-citations. Please revise accordingly.

  • To provide appropriate clinical context for the findings, the discussion around the potential of MRS as a non-invasive diagnostic tool—particularly its role in liquid biopsy—should be expanded. We strongly encourage the authors to integrate recent literature on this topic: PMID 37047088

In conclusion, the work is both original and timely. While the genetic contribution to endometriosis has been widely explored, this study is among the first to systematically quantify and model the epigenetic component through methylation risk scores (MRS). I am confident that the manuscript will be suitable for publication once these minor issues are addressed.

Author Response

Comments 1: The "Materials" section should be moved before the "Results", as it provides essential context needed to fully understand and interpret the findings.

Response 1: We appreciate the reviewer’s feedback regarding the manuscript structure. However, the current format follows the journal’s required template, and we are unable to modify it. As such we have presented results first but have made efforts to provide a brief overview of the methodology at the beginning of each result section. Figure 1 also serves to try and summarise the methodological approach to aid interpretation.

Section 2.2 - “To identify whether DNAm contributes to the variation in endometriosis status seen among participants, we estimated the proportion of variance in endometriosis status that can be captured by DNAm using omics residual maximum likelihood analyses (OREML), a residual maximum likelihood analysis that calculates the estimates from an omics relationship matrix (ORM) generated from DNAm beta values of the endometrium tissue samples.”

Section 2.4 - “The performance of a combined-risk-score classification model was evaluated to demonstrate if MRS can contribute any additional classification value to the current PRS.”

Section 2.6 - “Weightings used to compute the PRS in this study were developed from European ancestry cohorts. Therefore, the performance of PRS may be underestimated when applied to a multi-ancestry cohort. Hence, we aimed to verify our results by restricting the development and evaluation of MRS analyses to only European genetic ancestry participants.”

We have also made the following changes to enhance the clarity and coherence of the manuscript:

The purpose of section 2.1 Factors contributing to variation in endometrial DNAm (Results) was clarified on page 3, lines 101 to 104. “To identify and address confounders that contribute to variation in endometrial DNAm, a total of 908 samples retained following methylation quality control filtering were included in statistical tests to identify any associations between the potential covariates with endometriosis status and DNAm principal components (PCs).” Steps to address the confounders have been explained on page 3, lines 109 to 113 (Results). “Thus, age and institutions were used as covariates during the subsequent analyses. Additionally, genetic PCs were included as covariates during the development of MRS to account for the difference in genetic ancestry and population structure. Surrogate variable (SV) analysis was also conducted to remove batch effects and any hidden sources of variation that were not accounted for by the selected covariates.”

We have also added an explanation of why a leave-one-institute-out approach was used to interpret the results on page 5, lines 174 to 176 (Results section 2.3 MRS captures DNAm differences between endometriosis cases and controls). “This leave-one-institute-out approach ensures that the associations observed are not driven by a single institute and account for institute-specific sources of biases.” A brief methodology on how the MRS is developed has been explained as well at the start of section 2.3 “An outline of the MRS development and evaluation pipeline is illustrated in Figure 1 We estimated the effect sizes of DNAm probes on endometriosis via MLM-based omic association (MOA), Multi-component MLM-based association excluding the target (MOMENT) and best linear unbiased prediction (BLUP) using four different training sets, each excluding one of the four institutions (1- Centre for Inflammation Research, University of Edinburgh (CIR); 2- University of California San Francisco (UCSF); 3- Oxford Endometriosis CaRe Centre (Oxford); 4- Institute for Molecular Bioscience (IMB)). Covariates included age, institutions, menstrual cycle phase and genetic PCs.”

Comments 2, 4: Some of the citations (e.g., references 52, 55, 56, and 57) appear not to align well with the statements they are intended to support. We recommend reviewing these references to ensure consistency and relevance.; To provide appropriate clinical context for the findings, the discussion around the potential of MRS as a non-invasive diagnostic tool—particularly its role in liquid biopsy—should be expanded. We strongly encourage the authors to integrate recent literature on this topic: PMID 37047088

Response 2, 4: Thank you for pointing this out. We have checked the alignment of statements with references and removed or consolidated references where appropriate. We have retained some of the mentioned references as we believe they support the statements.

Reference 52 is retained to show the use of PRS in capturing the risk of endometriosis. Following is a quote from the paper “Our results suggest that a PRS captures an increased risk of all types of endometriosis rather than an increased risk for endometriosis in specific locations. Although the discriminative accuracy is not yet sufficient as a stand-alone clinical utility, our data demonstrate that genetics risk variants in form of a simple PRS may add significant new discriminatory value…… In the combined Danish cohort, a significantly increased PRS was found for cases (mean 0.28, SD 1.04) compared with controls (mean −0.16, SD 0.94, P = 1.9·10−11) (Figure 3A). The discriminative ability of the 14-SNP PRS was AUC = 0.64 (Supplementary Table 4).”

We have also expanded the discussion around the use of methylation as a non-invasive diagnostic tool, including mention of liquid biopsy. However, we aimed to avoid overinterpretation of the potential clinical implications, given the limited clinical utility of MRS in its current form. Instead, we focused on the value of methylation as a tool for investigating the effects of environmental exposures on disease, as this represents a crucial avenue for understanding endometriosis pathogenesis and identifying potential therapeutic targets. We appreciate the suggestion and have also incorporated relevant recent literature to provide additional context on page 12, lines 399 to 425.

Several observational studies have reported an association between environmental exposures and endometriosis [54-57]. DNAm serves as a valuable biomarker for environmental exposures [58], providing insights into how pollutants, diet, and other external factors contribute to disease pathogenesis. Identifying these influences could enhance our understanding of how epigenetic modifications regulate key biological mechanisms, including hormonal regulation, cellular proliferation, and inflammation. This knowledge may lead to the discovery of novel therapeutic targets, ultimately improving endometriosis management.

Epigenetic markers also show promise as biomarkers for disease detection and risk stratification alongside non-invasive approaches such as detecting microRNAs in liquid biopsy [59], autoantibodies in blood [60] or menstrual fluid analysis [61], and could facilitate early diagnosis and personalized risk assessment, enabling identification of high-risk individuals and informing targeted preventative and treatment strategies. However, further studies are needed to assess the utility of DNAm biomarkers for early diagnosis, given several known challenges such as tissue-specific DNAm patterns [62], the sensitivity of DNAm signals to external confounding environmental as well as technical factors, and the dynamic change in DNAm signals across the lifespan [63]. These issues will need to be addressed before DNAm biomarkers can be reliably implemented in clinical practice.

The study supports the hypothesis that DNAm influences endometriosis risk independently of genetic variation, emphasizing the importance of molecular techniques in studying non-genetic factors. Integrating MRS with PRS has demonstrated an improved classification performance, reinforcing the predictive utility of epigenetic factors beyond common genetic variation. These findings underscore the need for comprehensive epi-genetic studies to explore how environmental exposures contribute to endometriosis pathogenesis, paving the way for novel preventive and therapeutic approaches tailored to individual risk profiles.

Comments 3: The author Mortlock is self-cited five times, which exceeds the recommended limit of three self-citations. Please revise accordingly.

Response 3: Thank you for highlighting this. We are very active in this field, so many of our previous publications can provide context and support for interpretation. After revision, we have removed two of the following references and retained those that are the most relevant:

Reference 7 – Removed.

Reference 43 – Removed.

Reference 5 – This paper reported the largest GWAS study on endometriosis to date and has provided comprehensive evidence on the genetic basis of endometriosis.

Reference 6 – Polygenic risk score weightings were derived from this paper.

Reference 27 – Data used for analyses were obtained from this study.